# Measuring the diffusion of innovations with paragraph vector topic models

**David Lenz**📷*, **Peter Winker***

Department of Economics, Justus-Liebig-University, Gießen, Germany

* david.lenz@wirtschaft.uni-giessen.de (DL); peter.winker@wirtschaft.uni-giessen.de (PW)

**Data Availability Statement:** The data underlying the results presented in the study are available from HEISE NEWSTICKER ONLINE (https://www.heise.de/newsticker/). The authors received permission to work with the data. However, they have provided a python script so that users can

## Abstract

Measuring the diffusion of innovations from textual data sources besides patent data has not been studied extensively. However, early and accurate indicators of innovation and the recognition of trends in innovation are mandatory to successfully promote economic growth through technological progress via evidence-based policy making. In this study, we propose Paragraph Vector Topic Model (PVTM) and apply it to technology-related news articles to analyze innovation-related topics over time and gain insights regarding their diffusion process. PVTM represents documents in a semantic space, which has been shown to capture latent variables of the underlying documents, e.g., the latent topics. Clusters of documents in the semantic space can then be interpreted and transformed into meaningful topics by means of Gaussian mixture modeling. In using PVTM, we identify innovation-related topics from 170, 000 technology news articles published over a span of 20 years and gather insights about their diffusion state by measuring the topic importance in the corpus over time. Our results suggest that PVTM is a credible alternative to widely used topic models for the discovery of latent topics in (technology-related) news articles. An examination of three exemplary topics shows that innovation diffusion could be assessed using topic importance measures derived from PVTM. Thereby, we find that PVTM diffusion indicators for certain topics are Granger causal to Google Trend indices with matching search terms.

## Introduction

The rapidly growing amount of digital information provides novel data sources for economic analysis with regard to identifying and measuring innovation trends. To exploit this valuable information, there is a growing need for automated information retrieval from large text corpora [1]. Meanwhile, great progress has been made in machine learning and neural network theory leading to the emergence of new methods for extracting high-quality indicators from text. However, the opportunities created by the ongoing digitization have not yet been fully acknowledged nor have they been extensively studied in the economic literature. Following [2], who suggested that machine learning methods should be more widely known to and used by economists, more research that appropriately incorporates these new data sets and methods needs to be conducted in the field of innovation economics. For a few early exceptions, see the broader economic-use cases discussed below.

download the data: (https://gist.github.com/davidlenz/2f14b28ead67ee1522c2d5fcbf76a45e).

**Funding:** The German Federal Ministry of Education and Research provided funding for the research project (TOBI - Text Data BasedOutput Indicators as Base of a New Innovation Metric; funding ID: 16IFI001, Prof. Dr. Peter Winker) https://www.bmbf.de/en/index.html The funders had no role in study design, data collection and analysis, decision to publish, or preparation of the manuscript.

**Competing interests:** The authors have declared that no competing interests exist.

In an innovation context, working with textual data is a well-established practice. However, most studies have used patent registers as their main source of text data and have relied on more traditional methods to transform text into something readable by a computer.

A major part of the literature considers either patents, some specific aspects of innovation, or technology trends when investigating innovation. [3] is an early work that took into account not only the number of filed patents or frequency of citations but also the primary texts of the patents. The authors utilized text mining techniques to extract important keywords to build frequency vectors representing single patents. Subsequently, the authors performed a network analysis to visualize the relationships between patents and proposed measures for "importance," "newness," and "similarity" of the patents. Later studies such as [4], who considered patents involving a light emitting diode (LED) and wireless broadband fields, also conducted a network analysis. [5] proposed a semantic approach for patent classification that can be used in addition to the traditional USPC (United State Patent Classification) classes and also helps predict technology convergence. Recent work by [6] described an automated approach for patent landscaping, i.e. the process of finding patents related to a particular topic. Starting with a human-selected seed set of patents and expanding it through citations and class codes, semi-supervised machine learning is used to prune the expanded data set. [7] applied clustering techniques based on keyword analysis and community detection on scientific publications related to research in embryology to reconstruct the dynamics of the embryology research field and to understand early-warning signals prior to the occurring of some special events. [8] considered texts of approximately 170,000 awards from 2000 to 2011 that were included in the portfolio of the National Science Foundation (NSF). The authors were interested in measuring the inter-disciplinarity of the NSF portfolio and introduced an improved topic model based on latent Dirichlet allocation (LDA, [9]. See [10] for an introduction to probabilistic topic modeling including the LDA algorithm.).

Our approach differs from previous work mainly in two aspects, the data and the methodology. While most studies focus on patents, our study is the first to examine news articles as a way to measure the diffusion of innovations. Additionally, very few studies concerned with innovation research used embedding methods to represent text. For example, one study closely related to ours is [11]. The authors utilized patent fillings dating back to 1840 to estimate their novelty and significance by quantifying the impact on future technological innovations, using a time-aware term-weighting scheme based on term-frequency inverse-document-frequency (tf-idf) that the authors specifically constructed for this purpose. With this approach, the authors can capture technological evolution over a long time span, demonstrating the captured trends as strong predictors of productivity at various levels. In the same line, we tried to capture the diffusion of innovations by representing potential innovations as topics that can be found in news articles using topic models. Our study differed from [11] in the time scope (20 years for our data set vs 170 years) as well as the data source (news articles vs patents). Further, we relied upon text embedding methods to represent text, compared to the tf-idf scheme utilized by [11]. We are also more interested in the diffusion of innovations than technological progress as a whole, for which our data set is not extensive enough.

Regarding the methodology, in natural language processing (NLP), topic modeling describes a set of methods to extract the latent topics from a collection of documents. Topic models have been applied to extract stock topics from financial news with an application to predict abnormal returns [12], to categorize 8-k fillings and determine which topics are associated to abnormal returns [13], to measure the novelty of financial news [14] and to determine the key issues faced by firms [15].

Researchers have investigated the effect of increased transparency on monetary policy [16] and the adaptation of topics found in a Norwegian business newspaper to model the impact of

news on the business cycle [17]. Scientific discourse modeling is also within the scope of topic models. For example, [18] identified topics in the *Journal of Economics and Statistics* to study whether or not the scientific discussion of topics correlates with the actual development of economic key indicators. [19] examined how central bank communications affect real economic variables using topic modeling, and [20] employed LDA to model the topics in the *Journal of Economic History* between 1941 and 2016.

For many years, LDA has been the algorithm of choice for modeling latent topics in text corpora. However, LDA only describes the statistical relationships between words in a text corpus based on co-occurrence probabilities, which might not be the best feature representation for text [21]. Furthermore, LDA reportedly has long computation times, especially with large text corpora [22]. The interpretation of topics generated by LDA is not always straightforward, and the necessary mental effort to give meaning to the extracted words can be tedious and demanding work [23].

We propose a topic model architecture based on neural embedding methods that can generate meaningful and coherent topics as an alternative to LDA. In particular, we use Paragraph Vector (also known as Doc2Vec, [24]) to compute vector space representations of text documents and Gaussian mixture models (GaussMMs) to cluster the resulting document vectors into meaningful semantic topics. We call this combination of embedding and clustering Paragraph Vector Topic Modeling (PVTM). Our main methodological contribution is the way in which we combined these algorithms to extract latent topics from text corpora. A methodologically related approach to PVTM is from [25], who identified relevant articles for systematic reviews using Paragraph Vectors to construct vector representations of documents and then cluster the document vectors using k-means. The resulting cluster centroids were interpreted as latent topics, and the topic probabilities for the documents were constructed using the distance between cluster centroids and document vectors. In comparison, while also relying on Paragraph Vectors to construct vector representations of documents, we employed soft clustering via GaussMMs to directly assess topic memberships at the document level.

The applicability of our approach was demonstrated on a corpus of news articles from the German IT news publisher *Heise Medien* from the last 20 years. We compared our constructed topic measures with Google Trend indices (GTIs, https://trends.google.com/trends/). Google Trends measures the evolution of the frequency of search terms entered by users of the site over time. The results are set in relation to the total search volume and have been available in weekly resolution since the beginning of 2004 for the whole world or individual regions.

Our results suggest that PVTM is well suited for topic modeling this type of text data. Clearly, the topic probabilites generated by PVTM might serve as a proxy to measure the diffusion of certain types of innovations. In particular, it became apparent that the topic importance measures generated by PVTM Granger caused GTIs for matching search terms.

The remainder of this article is organized as follows. Section 1 details the methodological background of our analysis. The news ticker data set and the experimental design are reviewed in Section 2. In Section 3 we discuss our findings. Section 4 summarizes our results and describes avenues for future research.

## 1 Paragraph vector topic modeling

Section 1 introduces the PVTM methodology to generate topics and topic membership probabilities. PVTM relies upon the Paragraph Vector algorithm to generate document representations that are then clustered using GMMs. Our main methodological contribution is the way in which we combined these algorithms to extract latent topics from text corpora. Also, to the best of our knowledge very few studies have used embedding methods in an innovation

context, e.g. [6]. Paragraph Vector borrows the main ideas from Word2Vec, which is why it is useful to discuss the Word2Vec mechanics for encoding single words before detailing Paragraph Vector. In the last part of this section, we discuss the document clustering algorithm and provide intuition about the outputs yielded by PVTM.

### 1.1 Neural embeddings of words and documents

Neural network based embedding methods like Word2Vec [26, 27] play an increasingly vital role for encoding the semantic and syntactic meaning of words. Word2Vec builds low-dimensional dense vector space representations which encode the meaning of a word in a given context. As similar words tend to appear in similar contexts [28], encoding words based on their local context captures interesting properties in the resulting vectors, which have been shown to represent the way in which we use these words [29]. Intuitively, words that share many contexts are more similar than words that share fewer contexts. Vector calculations on the resulting word vectors yielded useful results, for example, $\mathbf{v}_{Paris} - \mathbf{v}_{France} + \mathbf{v}_{Italy} \approx \mathbf{v}_{Rome}$ [26], where $\mathbf{v}$ is the vector representation for a given word. These meaningful representations of words can be used as features in a variety of NLP tasks, including topic modeling.

Word2Vec comes in two architectural variants: Skip-Gram (SG) and Continuous Bag of Words. We discuss the SG architecture in more detail, as this is relevant for our application of the Doc2Vec model.

### Skip-Gram

During model training, the SG architecture iterates over a given text corpus in fixed-sized sliding windows and generates (*context—target*) word pairs, where $q$ target words are considered on both sides of the context word, which is the word in the middle. Assume the following 5-word window ($q = 2$): *Innovation is good for business*. The context word $w_c$ would be *good*, and the target words $w_{t-2}, w_{t-1}, w_{t+1}, w_{t+2}$ are *Innovation, is, for* and *business*. This results in 4 (*context—target*) pairs, (*good—Innovation*), (*good—is*), (*good—for*) and (*good—business*).

Given such pairs of context and target words resulting from the sliding window for a document, the construction of the vector representation is done as follows. First, each word in the vocabulary was represented as a one-hot encoded sparse vector $\mathbf{v}$ of size $V \times 1$, where $V$ is the number of different words in the vocabulary. In a one-hot encoding scheme, a 0 indicated the absence of a word, whereas a 1 indicated the presence of a word. Thus, $\mathbf{v}_{word}$ represented a sparse vector consisting of zeros and a single 1 for the row corresponding to the word.

Second, for each context word $\mathbf{v}_c$, a lower dimensional (denser) vector representation $\mathbf{d}_c$ was obtained using a projection matrix $\mathbf{W_c}$ of dimension $D \times V$, where $D \ll V$:

$$\mathbf{d}_c = \mathbf{W_c}\mathbf{v}_c. \tag{1}$$

Eq (1) extracts the column of $\mathbf{W}_c$ corresponding to the non-zero entry in $\mathbf{v}_c$ to use it as a vector representation $\mathbf{d}_c$ for word $v_c$ in a $D$-dimensional vector space.

The same procedure was repeated for each target word using a different projection (weight) matrix $\mathbf{W}_t$ of size $D \times V$ and the one-hot representation of the target word $\mathbf{v}_t$ resulting again in a $D$-dimensional vector $\mathbf{d}_t$, which is the vector representation of the target word:

$$\mathbf{d}_t = \mathbf{W_t}\mathbf{v}_t. \tag{2}$$

The probability of finding a target word $t$ close to the context word $c$ was modeled based on the similarity between the corresponding vectors $\mathbf{d}_t$ and $\mathbf{d}_c$ in the low dimensional vector space. The dot product of both vectors $s = \mathbf{d}'_t\mathbf{d}_c$ was used as a measure of similarity, which is closely related to the cosine similarity but also takes into account the lengths of both vectors.

Then, the probability that a target word $t$ is observed close to context word $c$ was defined making use of the softmax function [30]:

$$p(t|c) = \frac{e^s}{\sum e^s},$$

(3)

where the sum in the denominator is over all $V$ words in the text, i.e., all potential target words.

The projection matrices $\mathbf{W_c}$ and $\mathbf{W_t}$ were obtained by maximizing the joint (log-)probability of observing the target words for all context words in the corpus:

$$\sum_{j=1}^{R}\sum_{k=1}^{n_j}\sum_{l=\max(k-q,1),l\neq k}^{\min(k+q,d_j)} \log(p(\mathbf{v}_l^j|\mathbf{v}_k^j))$$

(4)

$C = \{D_1, \ldots, D_R\}$ was the corpus of $R$ news articles, where each article consisted of several words, i.e. $D_j = \{\mathbf{v}_1^j, \ldots, \mathbf{v}_{n_j}^j\}$. The number of words per article $D_j$ was denoted by $n_j$, and the size of the corpus was denoted by $R$. Thus, given an article $D_j$, $\mathbf{v}_k^j$ was the current context word used to predict target word $\mathbf{v}_l^j$ that fell in a given range around the context word. The size of the context window is denoted by $q$. Gradient descent was used to iteratively update the weights in $\mathbf{W}_c$ until some convergence criteria was met. After training the weights, $\mathbf{W}_c$ acts as a lookup table for the vector representations of words.

We used negative sampling and thereby sampled 5 negative examples for each positive word. With negative sampling, the SG algorithm compared the observed word-context pairs with randomly generated unobserved pairs and minimized the probability of the negative pairs while maximizing the probability of the actual word-context pairs. Negative sampling avoids calculating the softmax function for all possible words in the vocabulary and has been shown to speed up training and improve the resulting word vectors. [31] showed that the Word2Vec-SG architecture with negative sampling is equivalent to a weighted logistic principal component analysis.

## 1.2 Paragraph vector

The Paragraph Vector algorithm [24] expands the ideas of Word2Vec to longer pieces of texts. Instead of word vectors, document vectors are learned during the training process. The resulting vector representations have been shown to capture latent semantic properties of the text fragments, such as the underlying semantic topic of a document. The Paragraph Vectors algorithm also comes in two variants: Distributed Bag of Words (DBOW) and Distributed Memory (DM). In our experiments, we focused on the DBOW methodology, as it has been shown to produce slightly better results compared to DM. Although [24] reported that the DM architecture seems to perform better, subsequent research came to different conclusions [32].

DBOW builds upon the Word2Vec—SG architecture but replaces the center word with a unique document ID. Thus, instead of conditioning a single word on its surrounding words, the whole document is conditioned on the words appearing in it.

## 1.3 Gaussian mixture clustering

Topic modeling uses clusters of important words to define topics, where different topics may share some words. In our approach, this was done by clustering the vector representations obtained from Paragraph Vectors and then determining the most relevant words per cluster. A GaussMM (see, e.g., [33]) is a parametric probability density function represented as a weighted sum of Gaussian component densities [34]. GaussMMs employ the expectation

maximization algorithm [35] to fit a mixture of Gaussian models to a given data set and can be used to represent normally distributed subpopulations within an overall population.

GaussMMs have been used to track multiple objects in video sequences [36], to extract features from speech data [37] and for speaker verification [38]. Compared to frequently used clustering techniques such as k-means [39] or mean-shift [40], GaussMMs offer the advantage of soft clustering the data. Soft clustering allows multiple cluster memberships per document, so each document can be represented as a probability distribution over the cluster memberships. The result of the process is a matrix with one row per document and one column per identified cluster, where each entry represents the probability of belonging to a certain cluster. Given that Paragraph Vectors capture latent topics in the corpus, it is reasonable to suggest that clustering the resulting document vectors can be seen as identifying latent topics.

Particularly, given a $D$-dimensional vector representation $\mathbf{d}_r$ of a news article $D_r$ and a preset number of Gaussian components $M$ with mixture weights $w_i$, the GMM was defined as a weighted sum over the $M$ Gaussian components, where the mixture weights satisfied the constraint $\sum_{i=1}^{M} w_i = 1$:

$$p(\mathbf{d}_r|\lambda) = \sum_{i=1}^{M} w_i g(\mathbf{d}_r|\mu_i, \Sigma_i) \tag{5}$$

Thereby, each mixture component $g(\mathbf{d}_r|\mu_i, \Sigma_i)$, $i = 1, \cdots, M$ was defined as a D-variate Gaussian function of the form

$$g(\mathbf{d}_r|\mu_i, \Sigma_i) = \frac{1}{(2\pi)^{\frac{D}{2}}|\Sigma|^{\frac{1}{2}}} \exp\left\{-\frac{1}{2}(\mathbf{d}_r - \mu_i)^T \Sigma_1^{-1}(\mathbf{d}_r - \mu_i)\right\} \tag{6}$$

with $\mu_i$ and $\Sigma_i$ representing the mean vector and covariance matrix respectively, and $\lambda$ collecting all parameters of all mixture componentes, i.e.,

$$\lambda = \{w_i, \mu_i, \Sigma_i | i = 1, \cdots, M\} \tag{7}$$

The optimal parameter configuration $\lambda$ was estimated by iteratively updating the model components to best fit the training data using the EM algorithm. $p(\mathbf{W}_c|\lambda) = \Pi_{r=1}^{R} p(d_r|\lambda)$ is the GaussMM likelihood given the training data $W_c = (\mathbf{d}_1 \ldots \mathbf{d}_R)$. Starting with an initial configuration $\lambda_{old}$, a new configuration $\lambda_{new}$ was computed such that $p(\mathbf{W}_c|\lambda_{new}) \geq p(\mathbf{W}_c|\lambda_{old})$ for $R$ training vectors collected in $\mathbf{W}_c$. The initial configuration was computed using k-means, and the mixture components were updated according to

$$w_i^{new} = \frac{1}{R}\sum_{r=1}^{R} p(i|\mathbf{d}_r, \lambda_{old}) \tag{8}$$

$$\mu_i^{new} = \frac{\sum_{r=1}^{R} p(i|\mathbf{d}_r, \lambda_{old})\mathbf{d}_r}{\sum_{r=1}^{R} p(i|\mathbf{d}_r, \lambda_{old})} \tag{9}$$

$$\sigma_i^2 new = \frac{\sum_{r=1}^{R} p(i|\mathbf{d}_c^r, \lambda_{old})\mathbf{d}_r^2}{\sum_{r=1}^{R} p(i|\mathbf{d}_c^r, \lambda_{old})} - (\mu_i^{new})^2, \tag{10}$$

where (8) is the update for weight $w_i$, the means are updated according to (9) and (10) details the variance re-estimation, in this case, for a diagonal covariance. The a posteriori probability

for component $i$ is given by

$$p(i|\mathbf{d}_r, \lambda) = \frac{w_i g(\mathbf{d}_r|\mu_i, \Sigma_i)}{\sum_{i=1}^{M} w_i g(\mathbf{d}_r|\mu_i, \Sigma_i)} \tag{11}$$

The downside of GaussMMs is that the number of mixture components $M$ needs to be specified beforehand, and the algorithm is always going to use all $M$ components. This gives rise to the need of external validation methods. One way of evaluating the quality of a given GaussMM clustering is to use theoretical criteria like the Bayesian information criterion (BIC, [41]), which is the approach we took.

The procedure described in 1.1—1.3 is referred to as PVTM. Similar to LDA, the most widely used topic model, results yielded by PVTM are twofold: 1) a list of topics, wherein each topic is associated to certain words especially relevant in the context of the topic, and 2) a document-topic matrix, where each document is assigned a probability for each of the topics determined in 1) to belong to this document. Topics were found by clustering document embeddings obtained from Doc2Vec.

The document-topic matrix is created by applying the GaussMM with optimized weights to the vector representations of the documents. The result was a probability distribution across all topics (clusters). The most important words per topic were determined by the proximity of the trained word vectors to the topic vector. While learning the document embeddings, word vectors were also trained and embedded in the same space. The cosine similarity could then be used to determine the most similar words to the cluster centers (= topic vectors). These words then formed the word list for the corresponding topic. The topic vectors corresponded to GaussMM cluster centers. A disadvantage of the method is that a large corpus is necessary to get reasonable word vectors (and also document vectors).

The intuition behind PVTM is that Doc2Vec embeddings group similar documents into similar regions of the embedding space. Using a clustering algorithm on the high dimensional document representations yields document clusters, i.e., regions with a high density of documents sharing similar latent elements. PVTM makes these latent elements accessible in the form of words located in close proximity to the topic.

## 2 Discovering innovation-related topics from news articles

We applied PVTM to news articles in an effort to discover innovation-related topics and measure their diffusion by means of topic probabilities over time.

### 2.1 Technology-related news corpus

The data set was formed by news articles published by the German IT news ticker *heise online* (https://www.heise.de/newsticker/) in their news-ticker archive from 1997 to 2016. The total number of news articles was 174, 532, resulting in an average of 8, 727 articles per year. However, the number of articles per year before the 2, 000s was considerably lower compared to subsequent periods. The average news article consisted of 278 words. Fig 1 details the number of documents per year and the number of words per document.

We choose this corpus for the following reasons. First, the corpus has a clear technology reference, which makes the identification of technology-related topics easier. Second, the corpus is available over a long period of time, 20 years, making it possible to measure long-term processes such as the diffusion of innovations in the first place. Third, news are available in a timely manner in comparison to, for example, patent data so that adjustments or trends can be identified in the short term, which is an important criterion for Science, Technology and Innovation (STI) policy. One of the potential disadvantages is that the corpus is in German, which

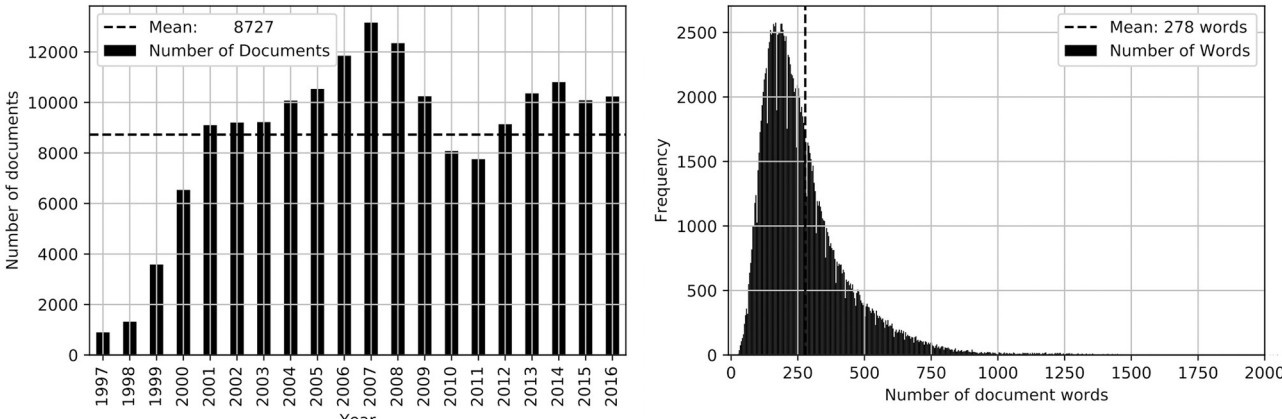

**Fig 1. Text corpus descriptive statistics.** A: Number of documents per year. B: Distribution of the number of words per document.

makes the results less accessible to non-German-speaking readers. This disadvantage is partially mitigated by the fact that many of the frequently used terms in current technologies spring from English and can therefore be understood by non-German-speaking readers. Apart from that, our question was specifically aimed at measuring the diffusion of innovations via news in one country. Germany is an example of an industrialized country. Future studies could transfer these ideas to other countries.

## 2.2 Data preprocessing & parameter settings

We removed all non-alphanumeric characters and lowercased the resulting words. Next, we applied popularity-based pre-filtering, which is a commonly applied technique in recommendation systems [42]. This can be seen as removing corpus-specific stop words from the vocabulary by setting an upper and lower threshold, defining the number of documents in which a word is allowed to occur before it is considered to be too frequent or infrequent. All words that occured in more than 65% of documents and in less than 0.05% of documents were filtered, as these words appeared to be too common/rare to be considered useful for our application of PVTM given the size of the corpus. The thresholds have been determined empirically, i.e., they have been adjusted until all corpus-specific stop words were removed. Thereby, we found that the upper threshold was not very sensible to changes. However, the lower threshold was a somewhat sensible parameter for PVTM, and as for the more liberal lower thresholds it was oftentimes not easy to determine meaningful topics due to the way the topic words were obtained. The most similar words to a topic vector were used as topic words. When the lower threshold was very low, very rare words, which might even appear in a single document only, were mapped close to the vector of the document in which they appeared. Thus, with no restriction on the minimum appearance of a word, words that appeared very few times and were very specific happen to be close to the topic vector and were chosen as topic words.

The algorithm to compute vector respresentations from news articles was run for 10 epochs, where each epoch consisted of going over all articles once. Following the default settings, we chose the dimensionality of the document vectors to be 100.

We used the BIC to find the optimal number of Gaussian mixture components $K$ and the best approach to construct the covariance matrices $\Sigma_i$. We considered four methods to estimate the covariance matrix: *Full* used the full covariance matrix for each individual component, i.e., each cluster can experience any shape, while *Diagonal* only used the diagonal of the covariance matrix per component, resulting in cluster shapes that were orientated along the

coordinate axes. *Tied* used a genereal covariance matrix for all mixture components; therefore, all clusters had an identical shape. *Spherical* made use of a single variance per component instead of a covariance matrix, resulting in spherical cluster forms in higher dimensions. All possible combinations of $K$ and $\Sigma_i$ for $K \in \{50, 1000\}$ and $\Sigma_i \in \{Diagonal, Full, Spherical, Tied\}$ were tested.

From there, we used a two-step procedure to find the optimal model parameters. We first iterated over the parameter space of $K$ in steps of 50 and started from 50, i.e., 50, 100, . . ., 1000. At every step during the parameter optimization procedure, all four options to construct the covariance matrices were evaluated. The best result $K^*$ was then used to construct a smaller search space $K \in \{K^* \pm 50\}$, which was searched in steps of 5. The optimal number of Gaussian mixture components $K$ (= topics) was found to be 675 after this run, which we kept as the final number of clusters. Eventually, the covariance matrices $\Sigma_i$ of the GaussMM were constrained to be *Diagonal* as this resulted in the lowest BIC scores for the data set at hand.

## 3 Approaching the diffusion of innovations from related topics

The OSLO manual [43] defines innovation as the implementation of a new or significantly improved product or process. Diffusion can be described as the process by which an innovation is adopted through certain channels over time among the members of a social system [44]. The diffusion curve is often drawn as a hump shaped line as shown in Fig 2.

A diffusion consisted of four main elements: (1) an innovation (2) that is communicated through certain channels (3) over time (4) among the members of a social system. Thereby, mass media outlets are effective tools to disseminate knowledge about innovations [44]. In our approach, the communication channel is the internet via a media outlet and the social system consisted of the users of the media outlet.

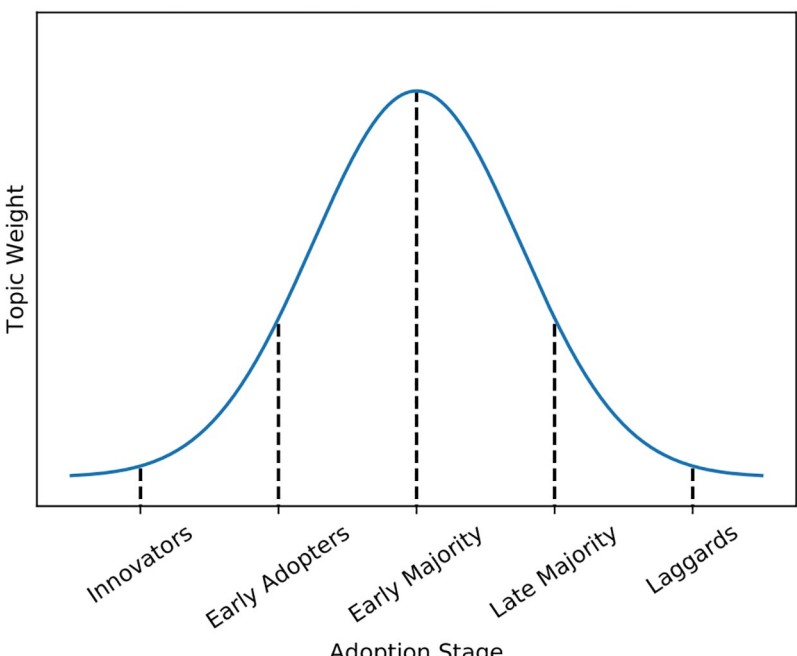

**Fig 2. The diffusion of innovations.** Adopters were categorized as innovators, early adopters, early majority, late majority and laggards depending on their adoption time.

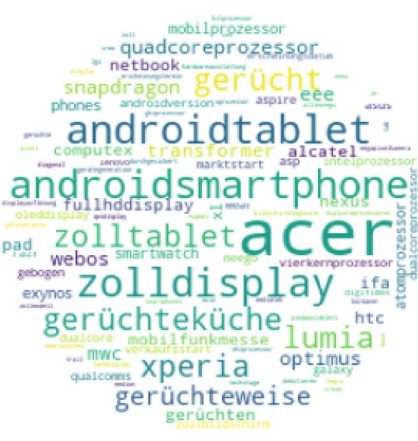
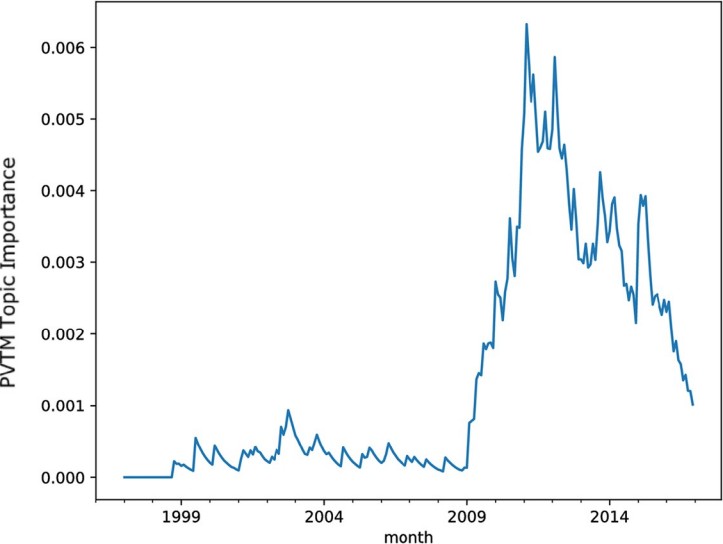

**Fig 3. Topic tablet.** A: Most relevant words. B: Monthly observations of the topic importance over time, smoothed using a 12-month exponentially weighted moving average.

Using topic modeling, we attempted to measure the diffusion of topics related to innovations by aggregating topic probabilities in a large technology-related news corpus over time. This way, conclusions about an innovation regarding adoption time and status could be drawn based on the aggregated probabilities for this topic in the corpus. We present three exemplary topics that were related to innovative activities. In particular, we used the evolution of their weights over time as an approximation of the diffusion curve of topic relevance.

The topics related to innovative activities are visualized in Figs 3–5. Thereby, the TABLET topic in Fig 3 represents a product innovation, the WIKIPEDIA topic in Fig 4 refers to a process innovation and the VIRTUAL REALITY topic in Fig 5 can be described as technology innovation.

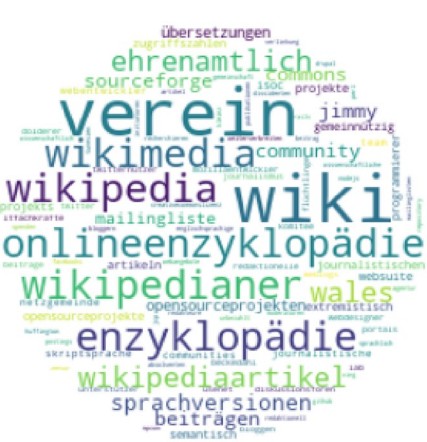
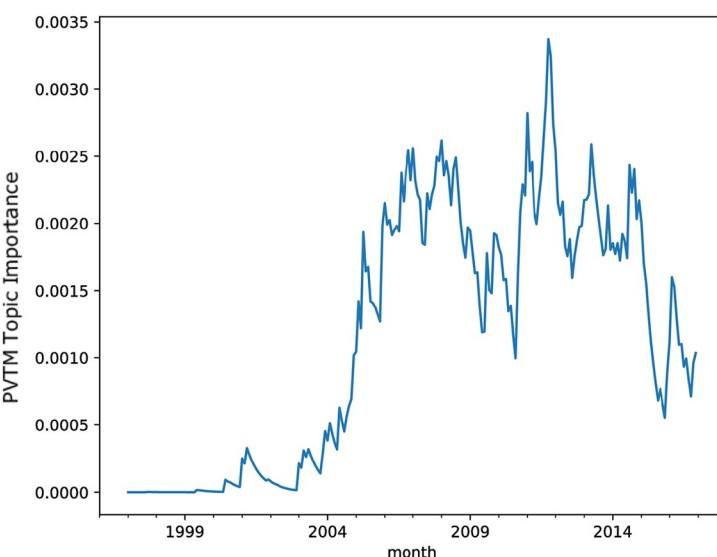

**Fig 4. Topic wikipedia.** A: Most relevant words. B: Monthly observations of the topic importance over time, smoothed using a 12-month exponentially weighted moving average.

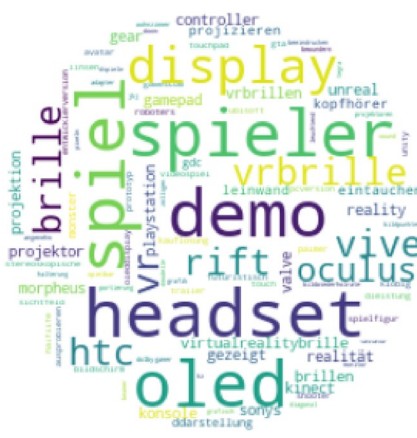

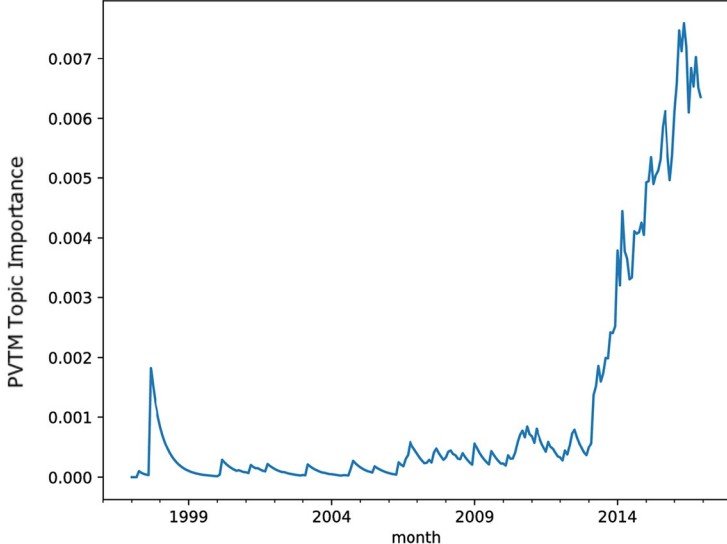

**Fig 5. Topic virtual reality.** A: Most relevant words. B: Monthly observations of the topic importance over time, smoothed using a 12-month exponentially weighted moving average.

Each figure consists of two panels. The left panel exhibits a wordcloud representing the most relevant words in the topic excluding stopwords. To this end we used stopwords from the python package stopwords (https://pypi.python.org/pypi/stop-words), added stopwords from the Natural Language Toolkit [45] and also included a list of common stopwords from https://github.com/6/stopwords-json.

The right panel in each figure shows a line plot of the evolution of topic weights over time. Given a certain time interval, we quantified the probability that a topic appears in the text corpus. Given a text corpus $C$ at a certain time frame—we considered one month—$t$, i.e., $C_t$, the probability for a single topic $p(T_i|C_t)$ was defined as the sum of the topic probabilities for all articles in the corpus in that time frame, weighted by the overall number of articles in period $t$, $D_t$.

$$p(T_i|C_t) = \frac{\sum_{d \in C_t} p(T_i|d)}{D_t} \qquad (12)$$

The first of the exemplary topics represented by the wordcloud in Fig 3 appears to be about (ANDROID)—TABLETS, as ANDROIDTABLET or ZOLL ("inch") are among the top words for this topic. This topic received major attention and started its take-off around 2006/2007 before peaking in 2010–2012, followed by a steep decline indicating a successful diffusion. Since then, the importance of the topic in the corpus has been decreasing and, given the topic timeline, one could conclude the adoption to the tablet technology is almost completed. The evolution of the topic's importance over time resembles a prototypical diffusion curve, spanning about 10 years.

In Fig 4 the WIKIPEDIA topic is detailed, which has been labeled according to top words such as WIKI, ENZYKLOPAEDIE ("encyclopedia") or WIKIMEDIA. The peak appears around 2006. Furthermore, the increase in the topic weights from 2002 to 2006 appears steeper than in the TABLET topic, implying a faster rate of diffusion. Similarly to the TABLET topic, diffusion can be considered as almost complete, which aligns with intuition. The observation of a rather fast diffusion rate of the WIKIPEDIA topic represents one of the advantages of the topic model based approach

to determine the diffusion state of innovations as it allows for a quite granular and timely approximation of this indicator. For the application, a steeper curve might correspond to a more disruptive innovation compared to innovations with flatter curves, as adoption is quicker. Starting with this observation, one might build a measure for the disruptiveness of an innovation based on the rate of diffusion measured through the importance of a topic over time.

The third topic shown in Fig 5 is labeled the VIRTUAL REALITY topic. The relevant words in this topic include OCULUS, RIFT and VIVE. Considering the importance of the topic over time, it became clear that the topic VIRTUAL REALITY had not peaked yet. However, it appears to be close to its peak, considering the overall shape of the aggregated weights over time, which resembles a Gaussian probability curve close to its maximum. Therefore, based on visual inspection, one might expect that after reaching the peak in 2018 or 2019, the relevance of this topic in the news corpus would decrease.

To go one step beyond the visual inspection and intuitive reasoning regarding the information content of our topic-based innovation measures, we compared their evolution over time with GTIs, which have been used in the literature as a leading indicator for a variety of economic quantities [46], including the diffusion of product innovations [47, 48]. In particular, [47] found that GTI data indicated a change in the interest for a product or technology well before it is visible in the purchasing behavior. The rationale for using these indicators was that potential users, in a first step, will search for information on new products, possibly making use of a search engine. The GTI provides a measure of the relative popularity of a specific search term in Google's search engine over time. Unfortunately, it is not known in detail how these measures are constructed. The period of highest interest in the time span under consideration is set to 100, and the remaining observations are adjusted according to the relative frequency compared to the reference period. Thus, GTI provide information about the relative importance of a topic over time, but not compared with other topics.

For our application, we consider monthly observations for the three terms WIKIPEDIA, VIRTUAL REALITY, and TABLET, which have been obtained for the period between 01/2004 and 12/2016 in Germany. The approach can be extended to further topics, which are well described by one word or short expressions, as they are used in Google's search engine. Some more examples are provided in the supporting information section in S1–S3 and S4 Figs. Fig 6 provides a graphical comparison of the GTI with the corresponding monthly relevance measures from our PVTM implementation. Thereby, the PVTM based measures have been smoothed to

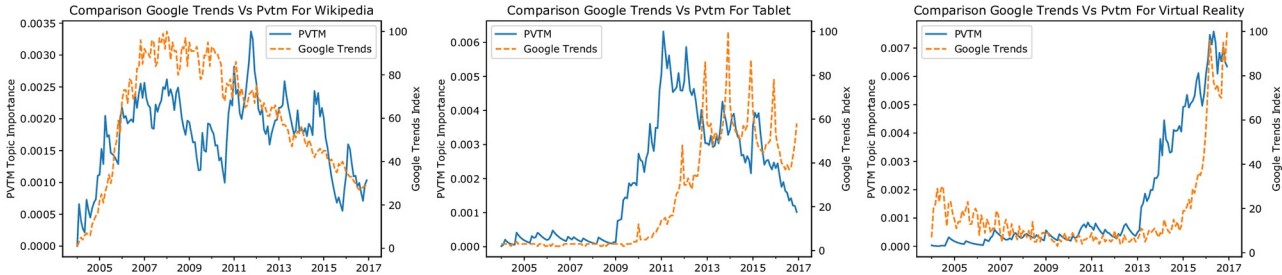

**Fig 6. Paragraph vector topic model diffusion indicators obtained from topic modeling compared to Google Trends indices.** A: WIKIPEDIA topic. B: TABLET topic. C: VIRTUAL REALITY topic. The PVTM measures are smoothed using the 12-month exponentially weighted moving average of the topic importance over time. We found statistical evidence that the unsmoothed PVTM topic importance measures Granger cause GTI for each of the presented topics.

reduce noise using an exponentially weighted moving average over the last 12 month. In each Fig, the left *y*-axis corresponds to the PVTM importance measure, while the scale of the GTI is given on the right *y*-axis.

The visual inspection of the plots in Fig 6 highlights some leading behaviour of the PVTM indicators as compared to GTI, which appears most pronounced for the TABLET topic. To substantiate these findings by means of a statistical analysis, we applied Granger causality tests [49, pp. 48ff]. Intuitively, this test analyzes whether past information of one series, e.g., from PVTM, helps to improve forecasts of another series, e.g., from GTI, beyond the information already contained in the past development of this series. A positive result of the test indicates that the first series provides additional information, which is leading the other series.

More formally, the Granger causality test is based on a bivariate vector autoregressive (VAR) model, in which the current value of each series is modeled by lagged values of both series under consideration. The null hypothesis of each test is that the joint impact of all lagged values of the other series has no relevant impact. The relevant test statistic asymptotically followed a chi-square distribution, if the series under consideration is stationary or cointegrated. Given that the visual inspection of the smoothed time series already indicated non-stationarity, we ran augmented Dickey-Fuller tests with automatic lag length selection based on the BIC assuming a maximum lag length of 13. As expected, the null hypothesis of non-stationarity could not be rejected for any of the variables under consideration at the 5 percent level. Consequently, Johansen's cointegration test was applied to identify potential long-run, cointegration relationships between the PVTM and corresponding GTI series. These tests indicate the existence of cointegration relationships. However, the original series are jagged and exhibit structural breaks which might lead to biased results of Johansen's test. Therefore, we decided to conduct the Granger causality test to first differences of the series as a more robust procedure even though we might lose some level information if the lag length is too small.

For each pair of variables, we selected the lag length of a VAR model in first differences based on the BIC with a maximum lag length of 24 month. The results including the lag length used are shown in Table 1.

The null hypothesis that PVTM does not Granger cause GTI may be rejected at the 1% significance level for VIRTUAL REALITY and TABLET and at the 10% significance level for WIKIPEDIA. This suggests that the topic indicators derived from PVTM contain leading information for the corresponding GTIs. One possible cause for this leading behaviour might be that news outlets and investigative journalists are likely to catch novel ideas faster than typical users of Google search such as consumers. Once reporting specific topics in news media increases, public interest also rises, possibly with some lag until people perceive and digest the news. Then, they

**Table 1. VAR Granger causality/block exogeneity wald test.**

| Topic | Null Hypothesis | Chi-sq | df | Prob. |
|---|---|---|---|---|
| Wikipedia | Δ PVTM does not Granger Cause Δ Google Trends | 5.565 | 2 | 0.062 |
| | Δ Google Trends does not Granger Δ Cause PVTM | 3.088 | 2 | 0.214 |
| Tablet | Δ PVTM does not Granger Cause Δ Google Trends | 40.127 | 13 | 0.000 |
| | Δ Google Trends does not Granger Cause Δ PVTM | 9.271 | 13 | 0.752 |
| Virtual Reality | Δ PVTM does not Granger Cause Δ Google Trends | 21.551 | 5 | 0.001 |
| | Δ Google Trends does not Granger Cause Δ PVTM | 15.309 | 5 | 0.001 |

use search engines like Google for further information about these technologies. Summarizing these findings, we suggest that importance measures derived from topic models might exhibit some leads compared to GTIs and, consequently, are more appropriate as early and leading indicators for the diffusion of innovations.

As a robustness check, we also ran the Granger causality tests in a cointegration setting as proposed by [50] and a test adding one extra lag to the VAR model to account for uncertainty regarding the level of integration and the existence of cointegration [49, p. 49]. These tests resulted in similar qualitative findings as the Granger causality tests applied to first differences reported here. As a further robustness check, we compared the topic importance of the WIKIPE-DIA topic to the number of new wikipedians per month in Germany. The results can be found in the supporting information section in S5 Fig and generally support the conclusion that PVTM generates sensible diffusion indicators.

## 4 Conclusion and outlook

There is an increasing interest in topic modeling, which is driven and ignited by the fast growing amount of textual data sources. In NLP, neural embedding methods have been shown to outperform standard methods on many tasks. They are therefore viable candidates for information retrieval from big text corpora for, for example, topic modeling.

We proposed PVTM, which uses Paragraph Vectors to construct document vectors and Gaussian mixture clustering to cluster the resulting vectors into meaningful topics. The applicability of our approach has been demonstrated by the emergence of coherent topics from technology-related news articles. Thereby, it became apparent that PVTM offers a useful alternative to discover latent topics in text corpora.

Empirical examples derived from the application of PVTM to (technology) news ticker data demonstrate the potential relevance for innovation economics. In particular, we were interested in measuring the diffusion of innovations over time. It appeared that the importance measures from PVTM Granger caused GTIs for the presented topics, thus offering a promising early indicator for innovation diffusions. However, further research is necessary to better understand PVTM's potential for innovation economics. One open issue of the methodological approach is an inherent forward looking bias arising from estimating the model based on the whole corpus only once. While the weights entering the Granger causality tests are only based on past documents, a future extension of the approach should also allow for a recursive estimation of the model as in [11]. However, this is not part of this study. It remains a task for further research to derive methods for prediction of diffusion and for the assessment of entities involved in innovative activities with regard to their stage of technology adoption. Of particular relevance is also whether an innovation is successful, as communication without adoption does not make the innovation. Taking the ongoing digitization into account, early identification and measurement of innovations will become more important. Therefore, we plan to analyze to what extent the method is applicable also in a dynamic setting. From an economic point of view, it would be interesting to know which entities act as the main players in a topic. A simple possibility would be to count how often an entity has been mentioned. Going further, one could use entity-related sentiment analysis to determine whether entities have a positive or negative impact on the topic. By identifying the first mentions of companies, they could be classified into categories such as innovators, early adopters, early majority, late majority and laggards with respect to specific topics. Using live news feeds could offer the possibility to capture the diffusion of innovations with very little delay, while including more news sources might inherit the potential to cover a larger share of ongoing innovative activities.

## Supporting information

**S1 Fig. Topic kindle.** A: Most relevant words. B: Monthly observations of the topic importance over time, smoothed using a 12-month exponentially weighted moving average.
(TIF)

**S2 Fig. Topic nfc.** A: Most relevant words. B: Monthly observations of the topic importance over time, smoothed using a 12-month exponentially weighted moving average.
(TIF)

**S3 Fig. Topic tegra.** A: Most relevant words. B: Monthly observations of the topic importance over time, smoothed using a 12-month exponentially weighted moving average.
(TIF)

**S4 Fig. PVTM diffusion indicators obtained from topic modeling compared to Google Trend indices.** A: KINDLE topic. B: NFC topic. C: TEGRA topic. Monthly PVTM observations were smoothed using a 12-month exponentially weighted moving average.
(TIF)

**S5 Fig. Topic wikipedia vs new wikipedians in Germany.** Monthly observations of the topic importance over time, smoothed using a 12-month exponentially weighted moving average, compared against the number of new wikipedians in Germany.
(TIF)

## Acknowledgments

We are indebted to the Heise Medien GmbH & Co. KG for providing access to its archives and the permission to apply text mining methods on these data. We further like to thank Monika Schuhmacher for her invaluable comments. Additionally, we are indebted to the reviewers at PLOS One for their valuable comments, which helped to markedly improve this paper. We also want to thank Viktoriia Naboka for her valuable research support.

## Author Contributions

**Conceptualization:** David Lenz, Peter Winker.

**Data curation:** David Lenz.

**Formal analysis:** David Lenz, Peter Winker.

**Funding acquisition:** Peter Winker.

**Investigation:** David Lenz, Peter Winker.

**Methodology:** David Lenz, Peter Winker.

**Project administration:** Peter Winker.

**Resources:** David Lenz, Peter Winker.

**Software:** David Lenz.

**Supervision:** Peter Winker.

**Validation:** David Lenz, Peter Winker.

**Visualization:** David Lenz.

**Writing – original draft:** David Lenz, Peter Winker.

**Writing – review & editing:** David Lenz, Peter Winker.

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
