## [Decision Letter · Decision Letter 0]

25 Jul 2019

PONE-D-19-15216

Measuring the diffusion of innovations with Paragraph Vector topic models

PLOS ONE

Dear Mr. Lenz,

Thank you for submitting your manuscript to PLOS ONE. After careful consideration, we feel that it has merit but does not fully meet PLOS ONE’s publication criteria as it currently stands. Therefore, we invite you to submit a revised version of the manuscript that addresses the points raised during the review process.

We would appreciate receiving your revised manuscript by Sep 08 2019 11:59PM. To enhance the reproducibility of your results, we recommend that if applicable you deposit your laboratory protocols in protocols.io, where a protocol can be assigned its own identifier (DOI) such that it can be cited independently in the future. For instructions see: http://journals.plos.org/plosone/s/submission-guidelines#loc-laboratory-protocols

We look forward to receiving your revised manuscript.

Kind regards,

Diego Raphael Amancio

Academic Editor

PLOS ONE

Journal Requirements:

Reviewers' comments:

Reviewer's Responses to Questions

**Comments to the Author**

1. Is the manuscript technically sound, and do the data support the conclusions?

Reviewer #1: Partly

Reviewer #2: Yes

Reviewer #3: Yes

Reviewer #4: Yes

2. Has the statistical analysis been performed appropriately and rigorously? 

Reviewer #1: Yes

Reviewer #2: Yes

Reviewer #3: Yes

Reviewer #4: Yes

3. Have the authors made all data underlying the findings in their manuscript fully available?

Reviewer #1: No

Reviewer #2: Yes

Reviewer #3: No

Reviewer #4: No

4. Is the manuscript presented in an intelligible fashion and written in standard English?

Reviewer #1: No

Reviewer #2: Yes

Reviewer #3: Yes

Reviewer #4: Yes

5. Review Comments to the Author

Reviewer #1: Review: PONE-D-19-15216, “Measuring the diffusion of innovations with Paragraph Vector topic models”

What the paper does:

This article proposes to use newly developed Paragraph Vector algorithms together with a Gaussian mixture model to estimate technology related news topics from a large news corpus and measuring their diffusion across time. The derived innovation topics are shown to Granger cause (predict) Google trends indices with matching search terms.

General comments:

I find the paper interesting, but with large room for improvement.

The idea, to use news articles and their underlying topics to measure diffusion of innovation seems sensible and novel. The approach, using paragraph vectors coupled with a Gaussian mixture models is creative and well placed. Finally, most of the analysis seems well conducted and the presentation is reasonably clear.

However, I have two main objections with the paper:

1) First, I find the result section rather disappointing. It seems to me that they only focus on three handpicked dimensions of this very high dimensional problem. I would like to see a much richer presentation of the results from their analysis in this respect. E.g., results for a wide range of innovation terms. Maybe using actual patent data registers might be useful here?

2) Second, as the article is written now, it is somewhat unclear what the methodological contribution is. Is it the how the authors use the paragraph embedding technology together with the Gaussian clustering algorithm, or is it something more fundamental. This should be clarified better.

Specific comments:

i) I did not really understand the last sentence of the abstract

ii) The authors should cite and relate to the article by “Measuring Technological Innovation over the Long Run”, by Bryan Kelly, Dimitris Papanikolaou, Amit Seru, Matt Taddy (2018). How do your paper differ, how do you contribute etc. ?

iii) Line17 to 19: These are typical LDA assumptions that need not be general for all other types of existing topic models. Should be clarified.

iv) Line 30: LDA term is introduced without having used before and explained.

v) Paragraph starting on Line 32: It is somewhat unclear if you use or develop a new approach here. This should be made much more clear. Also relates to my main comment. If you mostly use, but in a new way, existing methods, I think much of the math and explanations in Sections 1.1, and 1.3 could be removed. Would rather focus on the intuition and what various input and output matrices(vectors) vectors actually represent. This is somewhat unclear to me.

vi) How do you handle the potential forward looking bias arising from estimating the textual model/clusters on the same data as you do the Granger causality tests? This should be addressed, and I do not think you comment on it or address it now, or?

vii) Line 76: Doc2Vec is mentioned, but not explained. Related to the point above and the articles methodological contribution.

viii) The skipgram model. Is it not easier to just say that this is a logistic regression with latent parameters (word embeddings) and explanatory variables (context vectors)? Solved using a simple two-layer neural network?

ix) Do you apply the standard negative sampling scheme when solving the model?

x) In Section 1.3, it is not exactly clear to me what the final output from all this is. Should perhaps give the reader some more intuition for this.

xi) Below equation 11. Correct the text. Error in formatting.

xii) Section2.1. I think you should explain better why this corpus is a good choice in the current setting (and potential extensions etc. )

xiii) Line 191. K is used before it is defined and explained.

xiv) I find Section 3 confusing. What is the point of this section? You start to talk about the results and figures, but then comes section 4. Likewise, if the innovations should follow the shape you illustrate, how can you test this? Should you test this?

xv) In my version of the paper I could not find a figure of the word clouds that are mentioned in the text.

xvi) Line 311-318 feels somewhat speculative.

xvii) I do not understand why you need to use Johansen Trace Tests for cointegration in this setting. Would it not be enough to just difference the data to get them “roughly” stationary, and then run a battery of predictive tests (for many innovation terms)?

References:

Bryan Kelly, Dimitris Papanikolaou, Amit Seru, Matt Taddy (2018), “Measuring Technological Innovation over the Long Run”, NBER.

Reviewer #2: The article describes a novel approach to measuring innovation. This approach consists of a combination of some well-known methods. By considering this framework, tests were executed by using real data sets. Additionally, to analyze the results, the authors compared the proposed index of importance with Google trends index.

The analysis seems to be well-founded, but some crucial issues have to be corrected, as follows:

1- You mentioned LDA in line 30, but did not define what LDA means. It would be better to write "Latent Dirichlet Allocation (LDA) ". Additionally, I think that you could cite a book that describes this method.

2- In Skip-Gram subsection, you used two different symbols to mean $v_c$.

3- In Subsection 1.2, you named a method as Word2Vec-SG. However, it is not clear the meaning of SG. Is it Skip-gram? Please, define it in the text.

4- There is a problem with a citation on the least line of Subsection 1.2.

5- In subsection 2.2, you used a variable $K$ that was not defined before.

6- When you use BIC, do you mean Bayesian Information Criterion? It is not clear in the text.

7- Google Trends is crucial to your analysis. SO, I believe that you shall write a sentence explaining the index in the introduction. Furthermore, on page 8, you added a footnote with the website address, but this footnote should be on the first page where you mention it for the first time.

8- You mentioned that there are wordclouds in Figures 3 to 5, but there are no such figures in the PDF document.

9- Figure 6 is not necessary because its information is already shown in Figures 3 to 5.

10- You described some well-known methods of your paper but did not explain about the Granger Causality Test (GCT). I think that it would be appropriate to describe GCT and the advantages of the method when employed as a forecast of a series. Furthermore, you shall better explain all the techniques that take part in the comparison between PVTM and GTI.

11- In lines 292 and 298, you wrote GT instead of GTI.

12- The results regarding Table 1 could be better explained.

13- The data underlying the results are available on a website written in German. So, it was not possible to check it.

Reviewer #3: The paper delivers a good contribution to the literature on applying the "Paragraph Vector Topic Model"(PVTM) to a data set or corpus of technolgy news articles. The PVT model is taken from the machine learning literature and adapted to the data set. The presentation of the PVTM is concise and clear.

To evaluate the information content of the extracted topics, a Granger causality test is performed. All data are found to be I(1), the Johansen test confirms cointegration. The author proceeds using the Toda-Yamamoto approach by referring to Lütkepohl's textbook. This is inappropriate. This approach was developed for the case where the degree of integration is not clearly esdtablished. For the case here, the "trick" of adding an extra lag and estimating the VAR in levels is inferior to estimating a error correction model and analysing short-term and long-term aspects of cointegration separately (see the textbook on time series econometrics by Wolters, Kichgaessner and Hassler in this issue) by checking weak exogeneity and F-Tests/ Wald tests. Another appropriate test could also be found in the frequency domain Granger causality test proposed by Breitung and Candelon or Breitung and Schreiber.

I recommend to redraft the section using an appropriate test approach.

Furthermore there are a few type-setting errors (e.g. when the BIC test is referenced).

Reviewer #4: I have made my review in a pdf that is attached to this report. Please refer to this document for further details

6. PLOS authors have the option to publish the peer review history of their article (what does this mean?). If published, this will include your full peer review and any attached files.

Reviewer #1: No

Reviewer #2: No

Reviewer #3: Yes: Ulrich Fritsche

Reviewer #4: No

---

## [Author Response · Author response to Decision Letter 0]

8 Oct 2019

Please see the "Response to Reviewers.pdf" file.

---

## [Decision Letter · Decision Letter 1]

19 Nov 2019

PONE-D-19-15216R1

Measuring the diffusion of innovations with Paragraph Vector topic models

PLOS ONE

Dear Mr. Lenz,

Thank you for submitting your manuscript to PLOS ONE. After careful consideration, we feel that it has merit but does not fully meet PLOS ONE’s publication criteria as it currently stands. Therefore, we invite you to submit a revised version of the manuscript that addresses the points raised during the review process.

Minor edits are still necessary, as pointed out by the reviewers.

We would appreciate receiving your revised manuscript by Jan 03 2020 11:59PM. To enhance the reproducibility of your results, we recommend that if applicable you deposit your laboratory protocols in protocols.io, where a protocol can be assigned its own identifier (DOI) such that it can be cited independently in the future. For instructions see: http://journals.plos.org/plosone/s/submission-guidelines#loc-laboratory-protocols

We look forward to receiving your revised manuscript.

Kind regards,

Diego Raphael Amancio

Academic Editor

PLOS ONE

Reviewers' comments:

Reviewer's Responses to Questions

**Comments to the Author**

1. If the authors have adequately addressed your comments raised in a previous round of review and you feel that this manuscript is now acceptable for publication, you may indicate that here to bypass the “Comments to the Author” section, enter your conflict of interest statement in the “Confidential to Editor” section, and submit your "Accept" recommendation.

Reviewer #1: (No Response)

Reviewer #2: All comments have been addressed

Reviewer #3: All comments have been addressed

Reviewer #4: All comments have been addressed

2. Is the manuscript technically sound, and do the data support the conclusions?

Reviewer #1: Yes

Reviewer #2: Yes

Reviewer #3: Yes

Reviewer #4: Yes

3. Has the statistical analysis been performed appropriately and rigorously? 

Reviewer #1: Yes

Reviewer #2: Yes

Reviewer #3: Yes

Reviewer #4: Yes

4. Have the authors made all data underlying the findings in their manuscript fully available?

Reviewer #1: Yes

Reviewer #2: Yes

Reviewer #3: Yes

Reviewer #4: Yes

5. Is the manuscript presented in an intelligible fashion and written in standard English?

Reviewer #1: Yes

Reviewer #2: Yes

Reviewer #3: Yes

Reviewer #4: Yes

6. Review Comments to the Author

Reviewer #1: I think the authors have adequately addressed my comments raised in the previous round.

I only have two minor comments left:

1) Three are still some typos in the text. I encourage the authors to fix these before publication. E.g., line 122

2) The authors should check the reference list. Some of the articles listed as working papers are now published. E.g., reference [17]: Vegard H. Larsen, Leif A. Thorsrud, The value of news for economic developments, Journal of Econometrics, Volume 210, Issue 1, 2019, Pages 203-218.

Reviewer #2: All the comments have been addressed in the new version. However, there is no label on the y-axis of the new figures.

Reviewer #3: The authors reacted to the recommendations appropriately. Specifically, a robustness check of the Granger non-causality tests was integrated. The paper adds to the literature sufficiently. The method is very interesting and appropriate for the purpose of the study. Congratulations.

Reviewer #4: Dear authors,

Thank you for your response. I am now satisfied with the current version of the revisions and the responses to my comments.

7. PLOS authors have the option to publish the peer review history of their article (what does this mean?). If published, this will include your full peer review and any attached files.

Reviewer #1: No

Reviewer #2: No

Reviewer #3: Yes: Ulrich Fritsche, Universität Hamburg

Reviewer #4: No

---

## [Author Response · Author response to Decision Letter 1]

27 Nov 2019

Response to Reviewer 1

I think the authors have adequately addressed my comments raised in the previous round.

I only have two minor comments left.

We are again indebted to the referee for the careful reading of our review paper and

his/her overall positive assessment. We will describe in detail below how we reacted to

the specific comments made while preparing our revised version.

1. There are still some typos in the text. I encourage the authors to fix these before

publication. E.g., line 122

We thank the referee for this comment. We have again proofread the

paper very carefully and are confident that no typos are left at this

stage.

2. The authors should check the reference list. Some of the articles listed as working papers are now published. E.g., reference [17]: Vegard H. Larsen, Leif A.

Thorsrud, The value of news for economic developments, Journal of Econometrics, Volume 210, Issue 1, 2019, Pages 203-218.

We thank the referee for pointing out that some articles have been

published in the meantime. We checked all unpublished paper and

corrected the citation where appropiate.

Response to Reviewer 2

All the comments have been addressed in the new version. However, there is no label

on the y-axis of the new figures.

We are again indebted to the referee for the careful reading of our review paper and

his/her overall positive assessment. We will describe in detail below how we reacted to

the specific comments made while preparing our revised version.

1. There is no label on the y-axis of the new figures.

We consent with the reviewer that y-axis should have labels and therefore labeled the y-axis in the relevant figures with ”PVTM Topic Importance”.

Response to Reviewer 3

The authors reacted to the recommendations appropriately. Specifically, a robustness

check of the Granger non-causality tests was integrated. The paper adds to the literature

sufficiently. The method is very interesting and appropriate for the purpose of the study.

Congratulations.

We are again indebted to the referee for the careful reading of our review paper and

his/her overall positive assessment.

Response to Reviewer 4

Dear authors,

Thank you for your response. I am now satisfied with the current version of the

revisions and the responses to my comments.

We are again indebted to the referee for the careful reading of our review paper and

his/her overall positive assessment.

---

## [Editor Report · Decision Letter 2]

5 Dec 2019

Measuring the diffusion of innovations with Paragraph Vector topic models

PONE-D-19-15216R2

Dear Dr. Lenz,

We are pleased to inform you that your manuscript has been judged scientifically suitable for publication and will be formally accepted for publication once it complies with all outstanding technical requirements.

With kind regards,

Diego Raphael Amancio

Academic Editor

PLOS ONE
---

## [Editor Report · Acceptance letter]

26 Dec 2019

PONE-D-19-15216R2 

Measuring the diffusion of innovations with Paragraph Vector topic models 

Dear Dr. Lenz:

I am pleased to inform you that your manuscript has been deemed suitable for publication in PLOS ONE. Congratulations! Your manuscript is now with our production department. 

With kind regards,

on behalf of

Dr. Diego Raphael Amancio 

Academic Editor

PLOS ONE